# Exploring the Relationship between Combined Household Housing and Transportation Costs and Regional Economic Activity in Virginia

**Thomas W. Sanchez**

Urban Affairs and Planning, Virginia Tech, Arlington, VA 22203, USA; tom.sanchez@vt.edu

**Abstract:** Transportation is the second-largest expenditure category for households, accounting for nearly 20 cents of every dollar spent annually across the U.S. Only housing costs exceed transportation, and combined they represent a substantial burden on households. A primary economic connection between housing and transportation costs is related to the tradeoffs that households make in terms of residential location and what they have left of their household budget to spend on other needs. Families are forced to spend thousands of dollars annually on owning and operating private vehicles, forego wealth creation, and the ability to enjoy other benefits of homeownership. This analysis examines combined housing and transportation costs at the state level to regional economic performance. It contributes to the literature by testing the geographic scope of household expenditure burdens at this scale. Along with previous literature, this analysis provides evidence about the connection between the local and regional economic vitality and the burden of the combined effects of housing and transport on households. Overall, the results suggest that, from 2008 to 2018, these household cost burdens were a function of economic activity, household characteristics, and location in the state of Virginia.

**Keywords:** household costs; housing; transportation; economic impacts; location efficiency

## 1. Introduction

Transportation is the second-largest expenditure category for households, accounting for nearly 20 cents of every dollar spent annually for households in the United States [1]. Only housing exceeds transportation, and combined they represent a substantial portion of local, state, and national economic activity. A primary economic connection between housing and transportation is related to the tradeoffs that households make in terms of residential location and what they have left of their household budget for other needs. For example, as families are forced to spend thousands of dollars annually on owning and operating private vehicles (which are rapidly depreciating assets), they have less money to invest in homeownership, hindering wealth creation and the ability to enjoy other benefits of homeownership.

This is the first analysis of its type to examine combined household housing and transportation costs to economic performance at the state level. The article contributes to the literature by testing the geographic scale of household expenditure burdens in the state of Virginia. This has implications for transportation infrastructure investments, housing development planning, and regional economic planning. Previous theory, research, and practice have treated housing and transportation costs separately despite a relatively long history acknowledging the trade-off between housing and transport costs in urban economics.

The following literature review focuses on the dynamics between housing and transportation, including the costs, benefits, and trade-offs. Theory and prior research are briefly reviewed to set the context for the analysis of household cost burdens using the U.S. Department of Housing and Urban Development's (HUD) Location Affordability Index (LAI) data for the state of Virginia. The objective of the analysis is to examine how

these expenditure burdens correlate with local and regional economic performance during a post-recession period. In particular, the analysis anticipates that economic well-being of counties will benefit households, not only in terms of income but also access to employment locations, therefore reducing cost burdens. The literature review is followed by a description of the methodology used to analyze the application of the HUD LAI. The results of the statistical analysis highlight the relationships between county-level economic performance and the associated levels of housing and transportation cost burdens on households throughout the state. The results are followed by overall conclusions and recommendations for further research.

### 1.1. Literature Review

Nearly 200 years ago, Von Thunen's agricultural land model demonstrated an explicit relationship between location and farm rents, which flow from profits. Transport costs for farm goods were reflected in land rents because location (time and distance from markets) was a primary determinant of profits. Many years later, Alonso [2] applied a similar conceptual framework to the urban land market where households seek to maximize profits (satisfaction) by balancing the cost and inconvenience of commuting with the desire for space and income. Alonso also concluded that the determination of urban rents and densities is more complicated than the agricultural model of the 1800s because distances are not as extensive and location decisions are also affected by urban goods that vary by geographic location. These concepts form the basis of the housing and transport trade-off, which is a long-studied theory in urban economics [3–6]. These traditional models assumed monocentric cities, but metropolitan areas and modern commute sheds are more complex, and therefore, the trade-off between them may not be that simple. Location theory considering the interaction between transport and land use has a range of perspectives for residential, employment, retail, service, and industrial activities [7,8]. These relationships may not be expressed by individual theories, although including land market dynamics may indirectly account for these interactions. The fact that firms and households maximize utility in decision-making is a common element among these theories.

Investments in urban transportation networks increase productivity, resulting in population and employment density changes through impacts on capital–land ratios, development, and changes in the urban land market [9]. These investments also impact the distribution of benefits (such as productivity gains) to property owners [10,11]. As transportation facilities generate varying levels of service, spatial patterns reflect variable benefits. A wide variety of benefits from public programs accrue in varying degrees based upon geographic location. As mentioned, these public goods and services increase the productivity of sites within urban areas [12].

Transport improvements typically produce reductions in travel costs, travel time, net income increases, increased convenience, and other desirable outcomes [13–16]. Historically, access to land has been the primary function of personal transportation facilities. A concept that encompasses the previously mentioned benefits. As transportation costs change, land rent gradients change through market processes that link access to land use potential [3,10]. Access provided by highways allows land to be utilized for a profit, thus increasing its demand conditions [11]. There is a transfer of benefit from road users to property owners and local governments in the form of increased revenue streams as a function of increased land values. Other external benefits of transportation investments include increased economic productivity from business activity and employment generation [17,18].

### 1.2. Economic Well-Being and Development

The economic benefits of household location efficiency are many. Improved location efficiencies are passed on to households, businesses, and governmental entities as cost savings. Access to amenities is expected to produce improved quality of life [19], health benefits [20], and wage growth [21]. A reduction in housing and transportation cost

burden is expected to affect the economic well-being of households. Residents living in location-efficient neighborhoods require less time and expense. Bernstein et al. [22] report that regions with significant public transportation investments yield financial benefits to households in the form of affordable transportation options. Reducing transportation and housing cost burdens also increases access to economic opportunities in the form of education, jobs, and overall cost of living. High-quality public transit has been found to increase employment rates in U.S. cities and increasing transport system options increases productivity and potential for economic development [23,24]. Affordable rental housing proximate to frequent public transportation increases low-income households' access to jobs without assuming the costs of owning automobiles [25].

Location efficiency is associated with household economic resilience, where households can better respond to unexpected financial circumstances such as fuel price increases, vehicle failures, and income losses; also, in some cases, it can reduce the chances of eviction or foreclosure. According to Rauterkus et al. [26], the chances of foreclosure increase with higher neighborhood vehicle ownership rates, after controlling for income. Kaza et al. [27] find that increasing neighborhood job diversity and local accessibility reduces mortgage default risks. Commute time [28] and median vehicle ownership rates [29] can also increase the probability of default. Neighborhoods that are more walkable have been associated with fewer foreclosures [30]. Location-efficient mortgages may be a very effective tool to leverage access to affordable housing in transit-rich large metropolitan areas [31]. Properties foreclosed due to defaulted mortgages have been shown to increase blight and criminal activity [32], negatively affect city tax revenues, and consume municipal resources when cities become responsible for the vacant properties [33]. High rates of foreclosed properties also have negative impacts on surrounding property values [34,35].

### 1.3. Employment and Higher Wages

The location efficiency of an area is an important factor in attracting skilled labor to a region. Related factors include wage levels and economic vitality, commuting costs, and overall housing affordability [36]. Multiple studies have shown that increasing access to employment centers throughout a region leads to better employment opportunities and increased earnings [37–39]. Affordable commuting options improve workers' ability to increase both short- and long-term earnings [38,40,41].

Access to affordable commuting options is a significant factor in economic mobility among low-income, working households [42]. Low-skilled labor in central cities relies on accessible public transit [43–45]. Location-inefficient areas are likely to have spatial mismatches, resulting in unemployment among low-income, undereducated city residents. For households that have high transportation and housing cost burdens, the spatial arrangement makes it challenging for them to reach jobs they are qualified for because the cost of commuting to the place of employment outweighs the marginal economic benefit of those particular jobs. When these individuals find themselves unemployed, "the region experiences a loss of productivity in the form of underutilized human capital, thus reducing economic growth" [46]. This issue is not limited to the U.S. In the U.K., researchers Mattiolli, Lucas, and Marsden [47] (2016) have identified what they call "'car related' economic stress" (CRES), which impacts households and translates into negative regional economic consequences.

### 1.4. Business Costs and Formation

In a national survey of over 300 companies, two-thirds of respondents believe that a shortage of accessible affordable housing "is having a negative impact on retaining qualified entry-level and mid-level employees" and more than half attribute a certain amount of employee turnover to high commuting costs [48]. Additionally, daily long commutes due to lack of accessible affordable housing may contribute to traffic congestion [49]. Therefore, congested roads reduce local business profitability by increasing costs and shrinking the area from which businesses can draw both customers and workers [50]. Cities that fail

to address congested roads "may find their competitive edges slipping away to more favorable locations" [51] (p. 38). Location-inefficient areas are also shown to have high public service costs, with higher tax burdens to support local demand. Higher tax burdens are disincentives to a firm location while also reducing the capital and income available within the region [46].

*1.5. Location Efficiency*

Location efficiency includes residential and commercial developments that maximize accessibility and affordability. Koschinsky and Talen [52] describe location-efficient areas as those where transportation costs are currently low or where recent public investments will make transportation more affordable in the near future. This means that these areas are close to good transit and public services, as well as good walking and cycling conditions and other features that reduce automobile dependency.

Increasing location affordability through the HUD's Housing Choice Voucher Program (HCV) has shown mixed results. Affordable neighborhoods with accessible public transport are often in distressed areas, but not all HCV recipients are able to locate qualified housing [53]. Research from Oregon by Tremoulet, Dann, and Adkins [54] finds that overall HCV households are more likely to be located in areas with "high levels of residential and employment density, lower levels of car ownership, higher connectivity, more frequent and closer transit access, and higher Walk Score" (p. 17). However, recipients of vouchers who moved out of urban areas tended to relocate to less location-efficient areas. This may indicate that voucher recipients are being edged out of location-efficient areas and that moving to suburbs may provide more affordable housing but also higher transportation costs.

*1.6. Housing Affordability = Housing + Transportation Costs*

Transportation costs have been an overlooked household expense, and as a result, households only realize the actual expense after deciding on a location. Hickey et al. [55] reported that housing and transportation costs increased faster than incomes since 1999. Transportation costs include all trips made by households as part of their daily routines. For car owners, this includes the full costs of autos—car payments, insurance, maintenance, and fuel. For transit riders, it includes fares and time costs. Housing costs for renters include rent and utilities; for homeowners, it includes mortgage payments, property taxes, insurance, utilities, and, where applicable, condominium fees or mobile home park costs [56].

In recognition of this cost burden, multiple studies have analyzed how transportation costs vary between different locations and then used this information combined with housing costs to determine the total household and transportation burden for households. The methods used in these studies are varied and have evolved, as has their geographic scale. Bernstein et al. [22] utilized Consumer Expenditure Data to obtain transportation and housing expenditures at the MSA level. Subsequent studies achieved greater geographic detail by utilizing census tract or block-level data. To do this, transportation expenditure is needed to be modeled at the household level. A model developed by the Center for Neighborhood Technology (CNT) and Center for Transit-Oriented Development (CTOD) uses U.S. Census, transit system, National Household Travel Survey (NHTS), and other data sources to estimate a household's auto use, auto ownership, and transit use at the census tract level for particular household sizes and incomes.

Several studies to date have analyzed locational costs as a function of both housing and transportation expenditures, primarily in terms of impacts on household financial well-being at the neighborhood scale [57]. It should be noted that, as with many census-type data issues, the limitations of the LAI data should be carefully considered. Of particular concern to this study was the validity of the level of aggregation. Ganning [58] examined the application of LAI data at the census tract and block group levels and reported significant errors. For this reason, county-level analysis was used to analyze the state of Virginia. The analysis reported in this article is the first to apply these data to an expanded geographic

scale to examine the relationship of household cost burdens concerning local and regional economic performance.

## 2. Materials and Methods

Drawing from the literature, it is expected that local or regional economic conditions will be reflected in the level of housing and transportation cost burden experienced by households. In particular, this study examines how household cost burdens relate to indicators of economic activity, including business formation, employment, and aggregate payroll. The analysis used the HUD's Location Affordability Index (LAI) and County Business Patterns data. The analysis included locational characteristics and indicators of business vitality to predict the household cost burdens (housing and transportation) that impact heavily upon a household's economic well-being.

A linear regression model was used to predict household cost burdens (combined housing and transportation costs modeled by the LAI) for all Virginia counties and independent cities. The dependent variable was the percentage of the total household budget devoted to these costs for median income households. The independent variables in the model included the change in the number of establishments (businesses), change in total employment, and change in total payrolls, all at the county level. These are the primary variables that were considered to test the relationship between economic vitality and the relationship with household-level cost burdens. Other independent variables used to control for demographic and geographic variation included the percentage of residents under the age of 18 years old, the percentage of African American residents, the percentage of residents with a bachelor's degree or above, percentage of households with internet service, and the land area (size) of the county (or independent city). Given that the time period being analyzed occurred during an economic recovery period, three models were estimated for 2008 to 2013, 2013 to 2018, and 2008 to 2018. The cost burden from the 2014 LAI was used as the dependent variable for the first and the cost burden from the 2018 LAI was used for the second two time periods.

### 2.1. HUD Location Affordability Index

The HUD's Location Affordability Index (LAI) provides estimates of the combined housing and transportation costs for households at a variety of geographic aggregations. The LAI grew out of the Center for Neighborhood Technology's H + T Index, to more accurately represent the costs associated with particular residential location choices. The LAI data incorporate several data sources at the local, state, and federal levels to account for comprehensive household housing costs as well as transport expenditures. More detail on these data can be found on the HUD LAI Portal at: http://www.locationaffordability. info/lai.aspx (accessed on 11 May 2021).

### 2.2. Time Period for Analysis

The economic downturn beginning around 2007 was a challenge for the whole country. The economic performance data used in this analysis represent the approximate period of recovery, by including indicators for the years 2008, 2013, and 2018. This analysis not only focuses on the relationship between household cost burdens and economic activity but also the relationship between cost burdens and economic resilience throughout the state of Virginia. The analysis includes two time periods for both the household burden data and regional economic indicators. By looking at these periods, it is possible to examine how Virginia counties recovered, after 2008 as well as whether the recovery was sustained into the period from 2013 to 2018. The change in economic performance from 2008 to 2013 and 2013 to 2018 were used as indicators of economic well-being expected to have household-level impacts, reflected in the cost burdens. The regression analysis was used to test this relationship.

## 3. Results

As of 2018, Virginia counties appear to still be emerging from the 2008 economic recession. The period from 2008 to 2013 was characterized by losses of businesses and total employment, with slight increases from 2013 to 2018, but overall losses (on average) in the 10 years from 2008 to 2018. At the same time, aggregate payrolls increased an average of approximately 20 percent (see descriptive statistics in Table 1). It should be noted that the activity reported here is all before the COVID-19 pandemic, which was another shock to state, national, and international economies in 2020 and 2021. Table 1 also shows that median income households in Virginia paid an estimated 56 to 58 percent of their household budgets for housing and transportation. The data suggest that there was no statistically significant change in household cost burden from 2014 to 2018.

**Table 1.** Descriptive statistics.

| Variable | $n$ | Min. | Max. | Mean | SD |
|---|---|---|---|---|---|
| Change in No. Establishments (2008–2013) | 131 | −25.00 | 13.26 | −6.59 | 6.08 |
| Change in Employment (2008–2013) | 131 | −50.04 | 44.71 | −6.53 | 11.42 |
| Change in Payroll (2008–2013) | 131 | −63.03 | 59.03 | 3.72 | 15.42 |
| Change in No. Establishments (2013–2018) | 131 | −14.23 | 27.57 | 1.29 | 6.80 |
| Change in Employment (2013–2018) | 131 | −24.90 | 36.75 | 4.68 | 11.57 |
| Change in Payroll (2013–2018) | 131 | −36.29 | 82.62 | 17.44 | 18.45 |
| Change in No. Establishments (2008–2018) | 131 | −26.42 | 36.70 | −5.30 | 9.86 |
| Change in Employment (2008–2018) | 131 | −38.48 | 62.91 | −2.10 | 17.00 |
| Change in Payroll (2008–2018) | 131 | −37.12 | 163.28 | 21.20 | 25.20 |
| Percent African American | 131 | 0.40 | 77.20 | 18.80 | 16.47 |
| Percent Residents Under 18 years | 131 | 8.80 | 28.30 | 19.95 | 3.31 |
| Percent Households with Internet | 131 | 48.20 | 94.00 | 73.40 | 10.05 |
| Percent Residents with Bachelor's Degree | 131 | 9.00 | 78.50 | 26.59 | 13.45 |
| Land Area (Sq. Mi.) | 131 | 2.00 | 968.90 | 299.21 | 227.46 |
| Household LAI 2014 | 131 | 39.15 | 78.41 | 55.88 | 7.23 |
| Household LAI 2018 | 131 | 41.34 | 78.23 | 57.76 | 8.76 |

The three regression models predicted between 54 and 62 percent of the variation in household cost burden (see Table 2). The indicators of economic vitality (change in the number of establishments, change in employment, and change in payroll) were not correlated with household cost burden in the 5 years from 2008 to 2013, while the coefficients for the percentage of households with internet service, percentage of persons with a bachelor's degree or higher, and percentage of African American residents were statistically significant and negatively correlated with household cost burden. In other words, the higher each of these values was, households experienced lower cost burdens for combined housing and transportation expenditures. In the 5 years from 2013 to 2018, an increase in the number of business establishments was correlated with lower household cost burdens, but the change in employment and total payrolls were not statistically significant (see Table 2).

**Table 2.** Regression results.

| Variable. | 2008–2013 | 2013–2018 | 2008–2018 |
|---|---|---|---|
| (Constant) | 89.032 | 95.842 | 92.904 |
| | (0.000) | (0.000) | (0.000) |
| Change in No. Establishments (2008–2013) | 0.065 | | |
| | (0.516) | | |
| Change in Employment (2008–2013) | 0.012 | | |
| | (0.860) | | |
| Change in Payroll (2008–2013) | −0.028 | | |
| | (0.565) | | |

**Table 2.** *Cont.*

| Variable. | 2008–2013 | 2013–2018 | 2008–2018 |
|---|---|---|---|
| Change in No. Establishments (2013–2018) | | −0.224 | |
| | | (0.011) | |
| Change in Employment (2013–2018) | | −0.092 | |
| | | (0.247) | |
| Change in Payroll (2013–2018) | | 0.047 | |
| | | (0.315) | |
| Change in No. Establishments (2008–2018) | | | −0.164 |
| | | | (0.037) |
| Change in Employment (2008–2018) | | | −0.047 |
| | | | (0.464) |
| Change in Payroll (2008–2018) | | | 0.021 |
| | | | (0.589) |
| Percent African American | −0.073 | −0.098 | −0.090 |
| | (0.015) | (0.004) | (0.007) |
| Percent Residents Under 18 years | −0.030 | 0.010 | 0.055 |
| | (0.855) | (0.954) | (0.770) |
| Percent Households with Internet | −0.384 | −0.464 | −0.476 |
| | (0.000) | (0.00) | (0.000) |
| Percent Residents with Bachelor's Degree | −0.127 | −0.124 | −0.125 |
| | (0.047) | (0.050) | (0.048) |
| Land Area (Sq. Mi.) | 0.003 | 0.003 | 0.003 |
| | (0.127) | (0.281) | (0.207) |
| Dependent Variable | LAI 2014 | LAI 2018 | LAI 2018 |
| $n$ | 131 | 131 | 131 |
| Adj $R^2$ | 0.535 | 0.618 | 0.607 |
| F | 19.721 | 27.247 | 26.063 |
| Signif. F | <0.0001 | <0.0001 | <0.0001 |
| *p*-value shown under each coefficient | | | |

## 4. Discussion

The results of the regression analysis suggest that for the 10-year period from 2008 to 2018, Virginia counties experienced variable rates of economic recovery, likely more slowly from 2008 to 2013 compared to 2013 to 2018. This is particularly true when comparing the more urban parts of the state, such as the Northern Virginia portion of the Washington, DC, metropolitan region, with rural areas such as southwest Virginia. The rural, nonmetropolitan counties that have relied on agriculture and mining over the past 50 years continue to struggle. Any significant economic shocks affecting the state or region have pronounced impacts on rural areas due to their economic and demographic fragility. Household cost burden is yet another indicator of these challenges.

Other factors that should be considered for smaller scale analyses include the effects of economic stimulus programs directed to states for improvement of transportation infrastructure. The State of Virginia received nearly $700 million from the American Recovery and Reinvestment Act of 2009 (ARRA) [59]. ARRA funding was used to address structurally deficient bridges, highway resurfacing, rail network enhancements, and congestion relief projects spread throughout the state. To date there has been no comprehensive analyses of how these investments improved economic activity or transportation accessibility. In a similar fashion, the Coronavirus Aid, Relief, and Economic Security (CARES) Act of 2020 seeks similar stimulus outcomes and are currently underway in Virginia and throughout the U.S. Additional research on the period during and following the COVID-19 pandemic may also shed light on these dynamics.

## 5. Conclusions

The growing literature suggests that the housing and transportation cost burdens negatively impacting households also have implications for local and regional economic

vitality. These high costs impact many households, especially those with lower incomes, by leaving little disposable income for savings, retirement, health care, and education. In absolute dollars, the difference between a household with a low percentage of income being spent on housing and transportation-related costs and those with a high percentage in Virginia is in the range of $7000 to $18,000 annually, depending on the household income level. This is a significant amount, especially for economically vulnerable households.

These impacts are expected to extend beyond the household to local and regional business activities. This is especially true during times of slow employment growth and downturns, such as that experienced during the Great Recession from 2007 to 2011. The results of this analysis suggest that, for certain periods, household cost burdens and regional economic recovery were statistically correlated. The expectation that healthy local economies will translate into economic benefits for households concerning disposable income will need to be further explored. These dynamics are important to consider regarding infrastructure investments, system design, and improvements as well. The same is true for housing development planning throughout states and regions, and that other locational costs, such as transportation, be explicitly part of regional planning processes.

The relationship between transportation costs, housing costs, and economic activity has been discussed for many years, but coordination appears to be a significant challenge [60]. The results of this analysis add another dimension to the evidence about these relationships with a focus on household expenditures and regional consequences. Plans that impact the costs of housing and transportation may have important implications for labor inputs to regional economic vitality. In addition, more research is needed that helps to distinguish between aggregate, county-level metrics and household-level dynamics. For example, the regression results suggested that counties with greater proportions of African American residents have lower household cost burdens; however, further analysis is needed to explain what this means at the household or individual levels with respect to the economic indicators used in this analysis.

The results presented here expand the geographic scope of analyses examining location efficiency for economic indicators for the post-recession period of 2008 to 2018 for the state of Virginia. While the dynamics of location efficiency have been demonstrated at the local level, these results point to the state-wide variation in both household cost burdens and economic performance. For the period examined, the data suggest that local and regional economic well-being may correspond with household economic well-being but not to the extent anticipated. Following on to this research is a need for future analyses. Analyzing subsequent periods to detect trends during a relatively stable state-wide economy (2018 to 2023) will be an important comparison to the 2008 to 2013 and 2013 to 2018 time periods. Incorporating a temporal dimension will help to shed light on causality, such as the regional economic impacts on households and the households' impacts on the regional economy.

**Funding:** This research was funded by the State of Virginia, Executive Order (EO) 32, advancing Virginia's Housing Policy.

**Institutional Review Board Statement:** Not Applicable.

**Informed Consent Statement:** Not Applicable.

**Data Availability Statement:** Not Applicable.

**Conflicts of Interest:** The author declares no conflict of interest.

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
