# Peer review of "Exploring the Relationship between Combined Household Housing and Transportation Costs and Regional Economic Activity in Virginia"

_land, doi:10.3390/land10070742_

Round 1

Reviewer 1 Report

The paper addresses a topic, the relationship between an areas economic vibrancy and the total cost of housing and transportation.  The results are not as strong as expected, but the results merit publication.

Literature review:  The paper uses the HUD Location Affordability Index but does not cite a piece that critiques the accuracy of the index. Joanna P. Ganning (2017) It’s Good but Is It Right? An Under-the-Hood View of the Location Affordability Index, Housing Policy Debate, 27:6, 807-824, DOI: 10.1080/10511482.2017.1312478

Author Response

Comment 1

Literature review:  The paper uses the HUD Location Affordability Index but does not cite a piece that critiques the accuracy of the index. Joanna P. Ganning (2017) It’s Good but Is It Right? An Under-the-Hood View of the Location Affordability Index, Housing Policy Debate, 27:6, 807-824, DOI: 10.1080/10511482.2017.1312478

Very good suggestion. The following was added at line 209, "It should be noted that as with many census-type data issues, the limitations of the LAI data should be carefully considered. Of particular concern to this study was the validity of the level of aggregation. Ganning [58] examined the application of LAI data at the census tract and block group levels and reported significant errors. "

Thank you for your comments.

Reviewer 2 Report

This paper explored the change in location affordability index at a county-level for two periods after the 2008 recession in Virginia (2008-2013, 2014-2018, and aggregated: 2008-2018). Given the loaming recession before us as we move into the COVID-19 pandemic recovery, I suspect this area of analysis will be one of interest for those focused on economic recovery. While the paper was well written and described, there are a few areas where I believe the paper would be improved, if addressed.    Post 2008, several states developed transportation-related infrastructure stimulus programs. Can you provide context about relevant stimulus or transportation improvements that may have influenced (a) the economy or (b) transportation accessibilities?   This analysis is constructed at an aggregate level. More work is needed to distinguish between the construction of county-level metrics and household-level expenses. For example, the regression results make it clear that counties with greater proportions of African American residents have significantly lower LAI on average. However, at a household level, we might actually see structurally different relationships between the change in establishments, jobs, or payroll with change in LAI for African American households, compared with white households, for example.    It seems odd that %Bachelor's degree coefficients for 2008-2013 and 2014-2018 were both significance and approximately -0.124/-0.127, but the aggregate period coefficient was muted to nearly half the effect size (not sig. and -0.075). I recommend double-checking whether this is correct.    For the regression results, can you provide the p-value range? While this may be a style requirement for the journal, p-values themselves provide a sense of strength of relationships including identifying marginal significance.   Misc.
  • Line 161, the page number for [51] has mixed brackets/parentheses
  • Line 310, formatting issue missing leading tab?

Author Response

Comment 1

"Post 2008, several states developed transportation-related infrastructure stimulus programs. Can you provide context about relevant stimulus or transportation improvements that may have influenced (a) the economy or (b) transportation accessibilities?" 

To address this the following was inserted at line 304. "Other factors that should be considered for smaller scale analyses include the effects of economic stimulus programs directed to states for improvement of transportation infrastructure. The State of Virginia received nearly $700 million from the American Recovery and Reinvestment Act of 2009 (ARRA) [59]. ARRA funding was used to address structurally deficient bridges, highway resurfacing, rail network enhancements, and congestion relief projects spread throughout the state. To date, there have been no comprehensive analyses of how these investments improved economic activity or transportation accessibilites. In a similar fashion, the Coronavirus Aid, Relief, and Economic Security (CARES) Act of 2020 seeks similar stimulus outcomes and is currently underway in Virginia and throughout the U.S." [59] refers to ARRA funding received by Virginia from the federal government for transportation-related stimulus."

Comment 2

"This analysis is constructed at an aggregate level. More work is needed to distinguish between the construction of county-level metrics and household-level expenses. For example, the regression results make it clear that counties with greater proportions of African American residents have significantly lower LAI on average. However, at a household level, we might actually see structurally different relationships between the change in establishments, jobs, or payroll with change in LAI for African American households, compared with white households, for example."

The following was added at line 339 related to limitations of the analysis: "In addition, more research is needed that helps to distinguish aggregate, county-level metrics and household level dynamics. For example, the regression results suggested that counties with greater proportions of African American residents have lower household cost burdens, however, further analysis is needed to explain what this means at the household or individual levels with respect to the economic indicators used in this analysis."

Comment 3

"It seems odd that %Bachelor's degree coefficients for 2008-2013 and 2014-2018 were both significance and approximately -0.124/-0.127, but the aggregate period coefficient was muted to nearly half the effect size (not sig. and -0.075). I recommend double-checking whether this is correct."

Very good catch. The significance level previously shown was left-over from an earlier iteration of the regression results and was missed during editing. The coefficient was significant and in the range of the 2008-2013 and 2014-2018 regressions.

Comment 4

"For the regression results, can you provide the p-value range? While this may be a style requirement for the journal, p-values themselves provide a sense of strength of relationships including identifying marginal significance." 

P-values were added to the regression results table. 

Comment 5

"Line 161, the page number for [51] has mixed brackets/parentheses AND Line 310, formatting issue missing leading tab?"

Both were corrected.

Thank you for your insightful comments.